

**A global function of climatic aridity accounts for soil moisture stress on carbon**
**assimilation**
Giulia Mengoli[1], Sandy P. Harrison[2,3], I. Colin Prentice[1,3]
1: Georgina Mace Centre for the Living Planet, Department of Life Sciences, Imperial
College London, Silwood Park Campus, Buckhurst Road, Ascot, SL5 7PY, UK
2: Department of Geography and Environmental Science, School of Archaeology, Geography
and Environmental Science (SAGES), University of Reading, Reading, RG6 6AH, UK
3: Ministry of Education Key Laboratory for Earth System Modelling, Department of Earth
System Science, Tsinghua University, Beijing 100084, China
*Correspondence to*: Giulia Mengoli (gmengoli@ic.ac.uk)
**Abstract**
The coupling between carbon uptake and water loss through stomata implies that gross primary
production (GPP) can be limited by soil water availability through reduced leaf area and/or
reduced stomatal conductance. Vegetation and land-surface models typically assume that GPP
is highest under well-watered conditions and apply a stress function to reduce GPP with
declining soil moisture below a critical threshold, which may be universal or prescribed by
vegetation type. It is unclear how well current schemes represent the water conservation
strategies of plants in different climates. Here eddy-covariance flux data are used to investigate
empirically how soil moisture influences the light-use efficiency (LUE) of GPP. Well-watered
GPP is estimated using the P model, a first-principles LUE model driven by atmospheric data
and remotely sensed green vegetation cover. Breakpoint regression is used to relate the daily
value of the ratio $\beta(\theta)$ (flux-derived GPP/modelled well-watered GPP) to soil moisture, which
is estimated using a generic water-balance model. Maximum LUE, even during wetter periods,
is shown to decline with increasing climatic aridity index (AI). The critical soil-moisture
threshold also declines with AI. Moreover, for any AI, there is a value of soil moisture at which
$\beta(\theta)$ is maximized, and this value declines with increasing AI. Thus, ecosystems adapted to
seasonally dry conditions use water more conservatively (relative to well-watered ecosystems)
when soil moisture is high, but maintain higher GPP when soil moisture is low. An empirical
non-linear function of AI expressing these relationships is derived by non-linear regression,
and used to generate a $\beta(\theta)$ function that provides a multiplier for well-watered GPP as
simulated by the P model. Substantially improved GPP simulation is shown during both
unstressed and water-stressed conditions, compared to the reference model version that ignores
soil-moisture stress, and to an earlier formulation in which maximum LUE was not reduced.
This scheme may provide a step towards better-founded representations of carbon-water cycle
coupling in vegetation and land-surface models.
**1 Introduction**
The tight coupling between carbon uptake and water loss via stomata (Cowan and Farquhar,
1977; Manzoni et al., 2011) implies that gross primary production (GPP) can be limited by





water availability through reduced vegetation cover and leaf area index, reduced stomatal
conductance, or both. The reduction in evapotranspiration under water stress causes increased
sensible heat flux, warming the atmosphere above the canopy, which in turn causes a further
reduction in transpiration and plant carbon uptake (Seneviratne et al., 2010; Gentine et al.,
2016; Grossiord et al., 2020). Thus, an understanding of how water stress impacts plant
function is critical for predicting both the carbon cycle and climate implications of increasing
drought (Gentine et al., 2019).

A GPP model based on eco-evolutionary optimality (EEO) theory, the P model (Wang et al.,
2017; Cai and Prentice, 2020; Stocker et al., 2020), captures the trade-off between $CO_2$ uptake
and water loss. It produces realistic estimates of the seasonal and diurnal cycles of GPP under
well-watered conditions as well as or better than more complex models, despite having far
fewer parameters (Stocker et al., 2020; Harrison et al., 2021; Mengoli et al., 2022). However,
it overestimates GPP in seasonally dry environments because although it accounts for the effect
of atmospheric dryness in reducing stomatal conductance, it does not account for any additional
impact of soil-moisture stress. Previous application of an empirical stress function to reduce
GPP from well-watered values under dry soil conditions (Stocker et al., 2020) produced only
a modest improvement in simulated GPP. Given the potential for EEO-based models to provide
robust representations of vegetation and land-surface exchanges with the atmosphere (Franklin
et al., 2020; Harrison et al., 2021; Mengoli et al., 2022), it is important to develop a well-
founded approach to implement soil moisture stress in an EEO context.
Most vegetation and land-surface models assume that GPP at any location is maximal under
well-watered conditions (Bonan, 2019) and apply a stress function to reduce GPP as a function
of declining soil moisture when a critical threshold of soil water availability is reached. This
threshold may be universal, or prescribed by vegetation type (e.g. Best et al., 2011; Bousetta
et al, 2013; Oleson et al., 2013).  However, in an analysis of the influence of soil moisture
stress on the evaporative fraction (EF, the fraction of available energy used for
evapotranspiration, of which transpiration is usually the largest component), Fu et al. (2021)
showed that the critical soil moisture threshold at which EF is reduced varies across biomes
and climates. Fu et al. (2022) further showed that climatic aridity controls both this threshold
(which occurs at lower soil moisture in drier climates) and the maximum EF under well-
watered conditions, with vegetation in more arid climates using water more sparingly when
soil moisture is high, but continuing to extract water at a similar rate down to a lower threshold
value of soil moisture.  Comparing grasslands and (dry) savannas, they also showed that the
EF response of grasslands yields higher annual GPP than if the same ecosystems adopted the
EF response of savannas, and *vice versa*. These findings are consistent with a shift from
isohydric to anisohydric stomatal regulation with increasing climatic aridity (McDowell, 2011;
Kumagai and Porporato, 2012; Konings and Gentine, 2017), and with the idea that stomatal
strategies might have the effect of maximizing carbon assimilation over the annual cycle.
In this paper, we analyse daily GPP derived from 67 eddy-covariance flux towers representing
a wide range of hydroclimates. We fit breakpoint regressions to account for the impact of soil
moisture ($\theta$) on light use efficiency (LUE), expressed as the ratio $\beta(\theta)$ of flux-derived GPP to
GPP as predicted by the P model for well-watered conditions. We analyse fitted values of both
the maximum $\beta(\theta)$ and the critical threshold of $\theta$ as non-linear functions of the climatic aridity
index (AI), defined as the ratio of annual potential evapotranspiration (PET) to annual
precipitation. These relationships are used to generate a family of $\beta(\theta)$ functions, dependent on
AI, which can serve as multipliers of the modelled, well-watered GPP. The performance of the



resulting model is compared with that of the uncorrected P model, and with a version that
applies the soil-moisture stress function previously developed by Stocker et al. (2020).
**2 Methods**
*2.1 The P model*
The P model is a LUE model based on eco-evolutionary optimality theory for the trade-off
between carbon uptake and water loss (Prentice et al., 2014) and the acclimation and/or
adaptation of leaf-level photosynthesis to environmental conditions (Wang et al., 2017). The
model is driven by air temperature, vapour pressure deficit (VPD), incident photosynthetic
photon flux density (PPFD), the fraction of incident PPFD absorbed by leaves (fAPAR),
elevation (to calculate atmospheric pressure) and the ambient partial pressure of carbon dioxide
($c_a$). The model distinguishes $C_3$ and $C_4$ photosynthesis but does not require specification or
parameterization of any further plant functional types. When driven by satellite-derived
fAPAR, it reproduces the seasonal cycle and interannual variability in GPP at flux sites from a
range of natural vegetation types as well as geographic variation in GPP (Wang et al., 2014;
Balzarolo et al., 2019; Stocker et al., 2020) and temporal trends in GPP at flux sites (Cai &
Prentice, 2020). The P model was modified by Mengoli et al. (2022) in order to simulate diurnal
cycles, separating the instantaneous responses of GPP (with photosynthetic parameters fixed
over the diurnal cycle) from the acclimation responses of those parameters on a time scale of
around two weeks. This modified model is used here to simulate daily GPP, as the daily sum
of GPP computed on half-hourly timesteps.
Given the known tendency of the P model to overestimate GPP under dry conditions, the FULL
configuration of the current standard P model Pv1.0 (Stocker et al., 2020) includes an empirical
water stress function (also based on eddy-covariance flux data) that approaches 1 at a threshold
value of $\theta$ ($\theta^*$), where $\theta$ is plant-available water expressed as a fraction of soil water-holding
capacity, and $\theta^*$ is set to 0.6. The function declines more steeply with decreasing $\theta$ in drier
climates, with climatic moisture quantified by an estimate of the ratio ($\alpha$) of actual
evapotranspiration (AET) to potential evapotranspiration (PET). This function is used in Pv1.0
(FULL) as a multiplier of the modelled, well-watered GPP.
*2.2 Flux tower data*
GPP and meteorological data at 67 flux tower sites (Supplementary Table 1) were obtained
from the FLUXNET2015 data set (Pastorello et al., 2020). We used GPP based on the daytime
partitioning method (Lasslop et al., 2010; Pastorello et al., 2020). FLUXNET2015 provides the
meteorological variables required to run the P model, including air temperature, VPD and
PPFD on a half-hourly timestep. However, it does not provide fAPAR. We obtained fAPAR at
each site from the data set produced by Stocker et al. (2020) from the MODIS MCD15A3H
Collection 6 data set (Myneni et al., 2015). The original data set has a spatial resolution of 500
m and a temporal resolution of four days. Stocker et al. (2020) filtered these data to remove
points where clouds were present and derived daily data by linear interpolation. We used a
subset of the sites from Stocker et al. (2020), chosen to cover the full range of aridity with no
major gaps. Meteorological data and MODIS data were not available for some sites/years, so
analyses and simulations were based on different years at different sites (Supplementary Table
1). We only used the half-hourly records from each of the selected sites where the quality
control flags indicated that the observations were "good".



*2.3 Calculation of the GPP reduction factor*
We calculated the ratio $\beta(\theta)$ between flux-derived and modelled, well-watered GPP at each site
and day. Our approach differs from that of Stocker et al. (2020) in three key respects. First, our
fitted stress function is allowed to take values < 1 under well-watered conditions. We thus
allow for the possibility that ecosystems adapted to arid climates use water more conservatively
even when soil moisture is abundant. Second, in order to ensure consistency of the soil moisture
calculation across sites, we calculate daily soil moisture using the Simple Process-led
Algorithms for Simulating Habitats (SPLASH) model (version 1: Davis et al., 2017) with
simulated soil moisture converted to relative soil water content ($\theta$) by dividing by the generic
bucket size in SPLASH (150 mm). Third, we use AI (the ratio of PET to annual precipitation)
rather than $\alpha$ as a climatological index, because of its wider use in the literature and because
its calculation is independent of the SPLASH model's estimation of AET.
*2.4 Breakpoint regression analysis*
We used breakpoint regression (Toms and Lesperance, 2003) to estimate the maximum level
of the $\beta(\theta)$ ratio under well-watered conditions, and the critical threshold below which the ratio
declines linearly towards the wilting point, at each site. Before this analysis, we removed values
of flux-derived GPP below the 5[th] percentile (which gave highly variable $\beta(\theta)$ ratios) and
observations with greater than the 99[th] percentile of $\theta$, which would otherwise have dominated
the regression at many well-watered sites. Preliminary analyses showed that the intercept was
generally close to zero and that imposing the constraint $\beta(0) = 0$ had little effect at the great
majority of sites (Supplementary Figure 1). We therefore imposed this constraint resulting in a
regression model with just two parameters, the maximum level of $\beta(\theta)$ (*y*) and the critical
threshold of $\theta$ ($\psi$):
$$\beta(\theta) \; = \; \min\,[y,\,(y/\psi) \times \theta] \hspace{6cm} (1)$$
The non-parametric Kruskal-Wallace test was used to determine whether there were significant
differences in fitted parameter values among aridity classes.
*2.5 Calculation of the aridity index*
The length of the meteorological records in FLUXNET2015 is too short to calculate a
climatological index at most sites. We therefore derived AI using climate data for a 20-year
period (2001-2020) from the CRU TS 4.06 gridded climate data set (Harris et al., 2020). We
obtained precipitation data directly from the CRU data set and calculated PET using
temperature, precipitation and cloud cover from this data set as inputs to SPLASH version 1
(Davis et al., 2017). Of the 67 selected sites, nine were classed as arid (AI > 5), 22 as semi-arid
(2 < AI < 5) and 36 as humid (AI < 2) (Table 1, Supplementary Table 1). We removed two
sites classified as arid (AU-Lox, AI = 6.32, and US-Wkg, AI = 6.34) and one classified as
semi-arid (AU-RDF, AI = 2.16), either because they were irrigated crops (AU-Lox, AU-RDF)
or because the presence of extensive wetlands indicate that they were groundwater-fed (US-
Wkg). The derivation of the stress function was thus eventually based on analysis of 64 sites.



*2.6 Dependencies of parameters on aridity*
The breakpoint regression yielded values of two parameters ($y$, $\psi$) for each of the 64 sites. We
fitted relationships for each parameter as functions of site AI using non-linear regression. Both
parameters were fitted with a power function:
$$\text{parameter} \; = \; \min\,[a.\text{AI}^b, \; 1] \hspace{4cm} (2)$$
where $b$ is expected to be negative. This function is bounded above in order to avoid potential
values > 1 in extremely wet sites, although none were present in the data set.
*2.7 Application*
Equations (1) and (2) determine a unique $\beta(\theta)$ function for each value of AI. This function was
applied as a multiplier of modelled GPP:
$$\text{GPP}_{\text{new}} \; = \; \text{GPP}_{\text{ww}} \times \beta(\theta) \hspace{4cm} (3)$$
where $\text{GPP}_{\text{new}}$ is the revised, soil-moisture corrected GPP, $\text{GPP}_{\text{ww}}$ is the GPP simulated by the
P model without soil-moisture correction, and $\beta(\theta)$ is given by equation (1) with parameter
values derived from equation (2) as a function of site AI. We compared the predictions of GPP
obtained using this new soil-moisture stress function to the uncorrected GPP, and with
predictions obtained using the implementation of soil-moisture stress in Pv1.0 at all of the flux-
tower sites, with meteorological data provided for the site in the FLUXNET2015 data set and
fAPAR data from Stocker et al. (2020). The goodness-of-fit between modelled and flux-
derived GPP at each site was quantified by the root mean squared error (RMSE).

**3 Results**

The response of LUE to water stress could be described by equation (1) (Figure 1,
Supplementary Figure 2). Both the maximum assimilation level and the critical threshold at
which soil moisture stress starts to impact LUE were found to vary with aridity. The maximum
assimilation level under well-watered conditions becomes progressively lower from humid
through semi-arid to arid sites (Figure 2). The difference between humid, semi-arid and arid
sites is significant. The critical threshold is also reduced, such that water stress sets in at higher
soil moisture in humid sites than in semi-arid or arid sites (Figure 2). This difference is also
significant. Moreover, the slope of the stress function below the critical threshold becomes
progressively steeper with increasing aridity. Thus, plants growing in more arid environments
have a lower maximum LUE overall, but sustain this level under drier soil conditions (Figure
3). These relationships were also evident when the intercept was not constrained to zero
(Supplementary Figure 3).





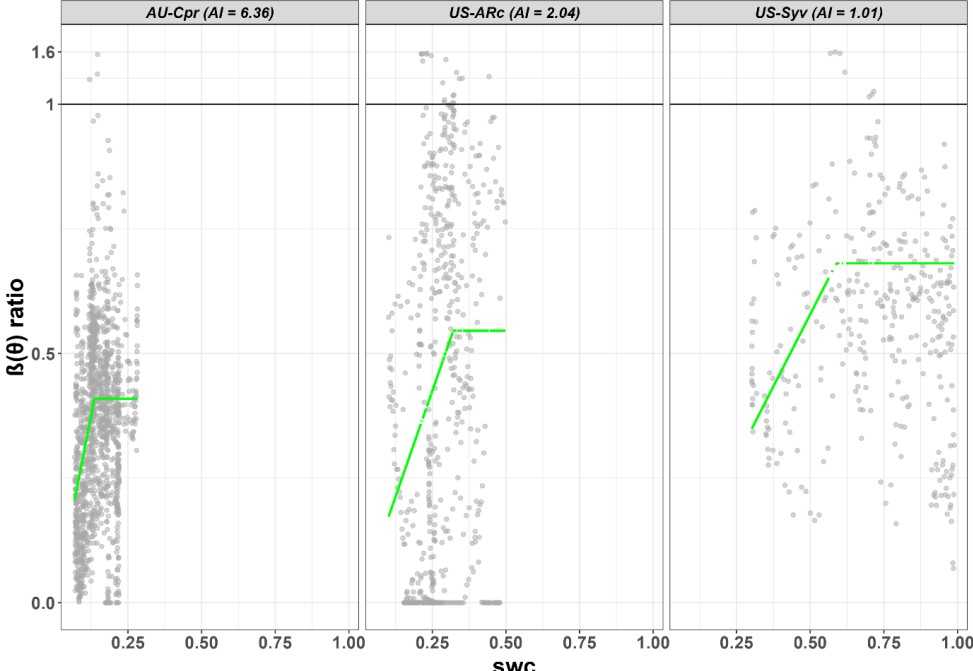

**Figure 1.** Examples of the fitted maximum β(θ) ratio (the ratio of actual flux-derived to modelled well-watered gross primary production) and its response to relative soil moisture below the critical threshold (green line) for three sites representing the range of climatological aridity levels. The β(θ) ratio and relative soil water content are both unitless. Note that the scale above 1 has been compressed for visualization purposes. Plots for all the sites used in the analysis are given in Supplementary Figure 2.





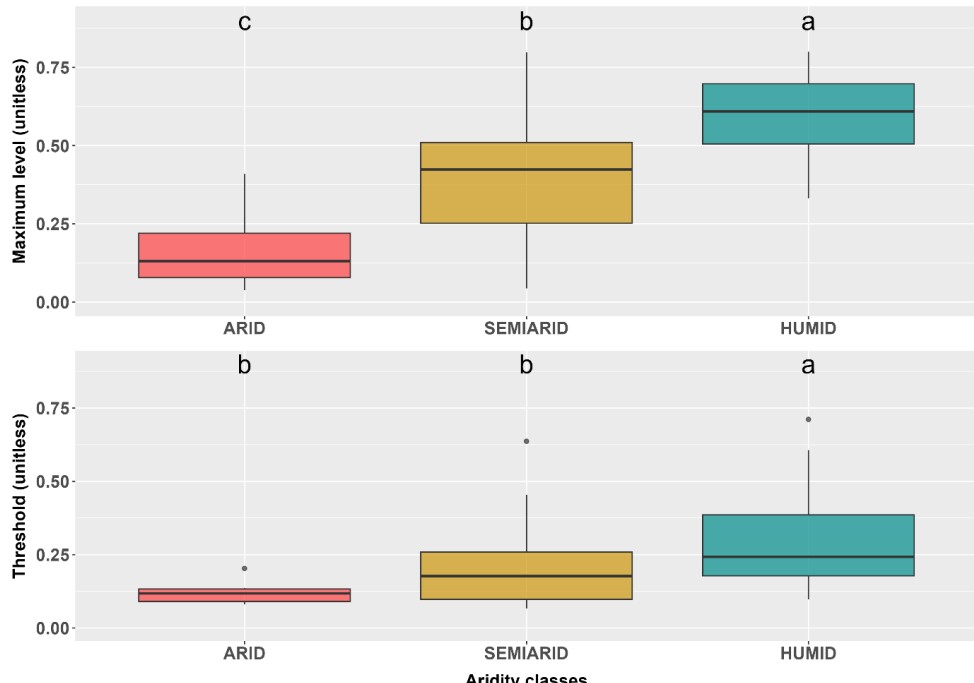

**Figure 2**. Box-plot comparison of the fitted maximum $\beta(\theta)$ ratio (the ratio of actual flux-derived to modelled well-watered gross primary production) (above) and the critical threshold value of soil moisture (below) under arid, semi-arid and humid conditions. Arid sites have AI > 5, semi-arid sites have AI between 2 and 5, and humid sites have AI < 2. The black line is the median, the box is the interquartile range and the whiskers show the range, with outliers shown as asterisks. Letters indicate whether the sets of values are significantly different based on the Kruskal-Wallace test.

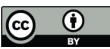


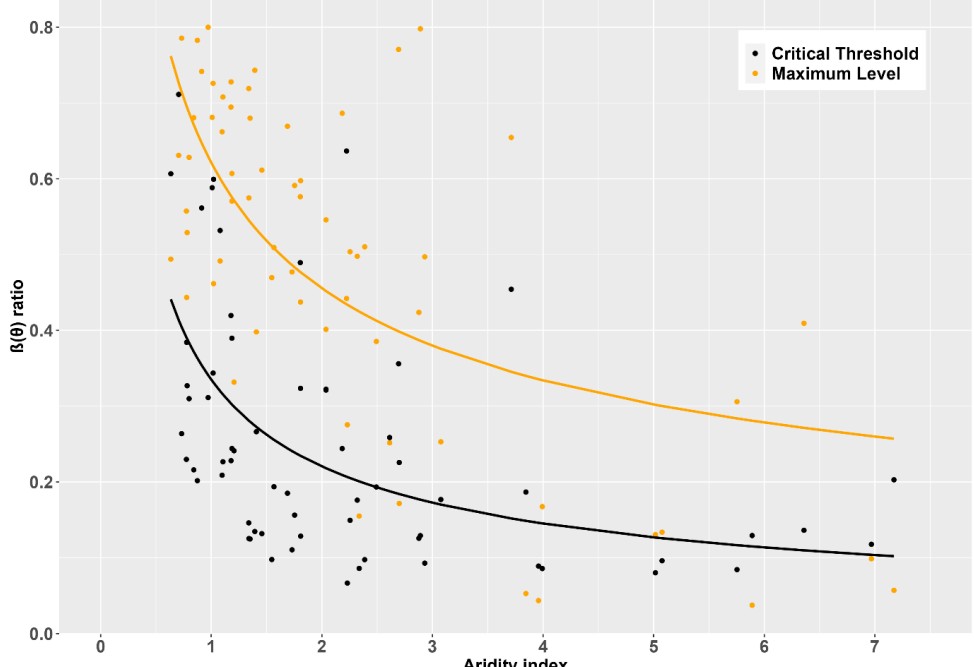

**Figure 3**. Values of the fitted maximum β(θ) ratio (the ratio of actual flux-derived to modelled
well-watered gross primary production) and the critical threshold value of soil moisture against
the climatic aridity index (AI), showing non-linear regressions of both parameters against AI.

Both model parameters showed non-linear relationships with AI that could be fitted using
equation (2) (Figure 4). Although there were some outliers, these do not seem to be related to
either vegetation type (Supplementary Figure 4) or the seasonal concentration of precipitation
(Supplementary Figure 5). The derived equations for the maximum β(θ) level (*y*) and the
critical threshold of θ (ψ) are as follows:
$y = \min [0.62 \ AI^{-0.45}, 1]$ (4)
and
$\psi = \min [0.34 \ AI^{-0.60}, 1]$ (5)



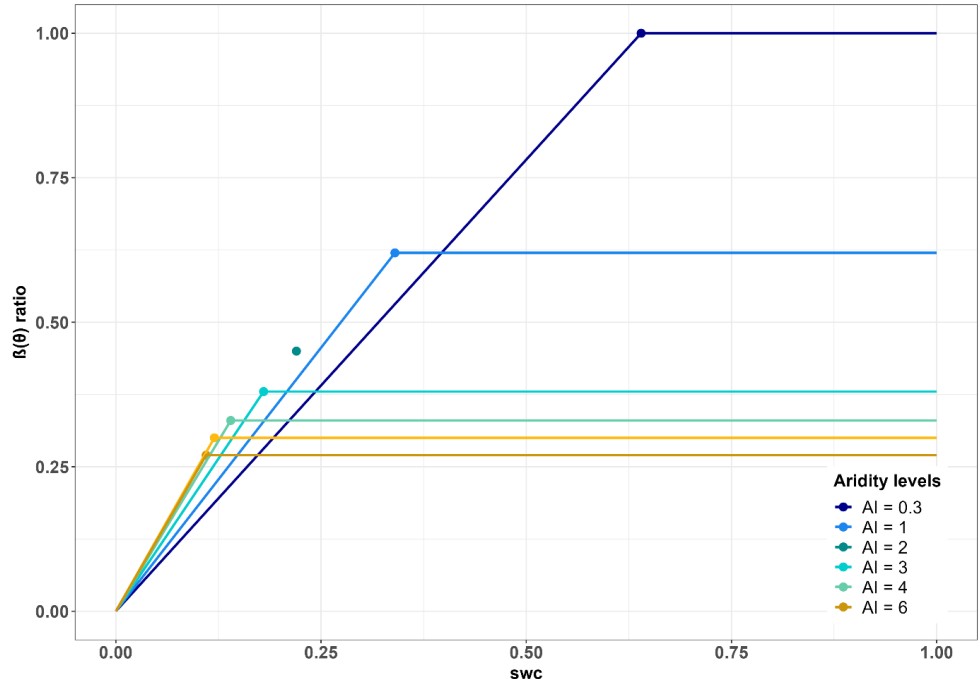

**Figure 4**. Predicted β(θ) ratio (the ratio of actual flux-derived to modelled well-watered gross primary production) functions based on the regressions shown in Figure 3, for different levels of the aridity index (AI).

Implementation of the new soil-moisture stress function produced a substantial improvement in model performance compared to simulations with no soil-moisture stress function (Figure 5, Supplementary Figures 6–8). At arid sites, simulations that did not account for soil-water stress overestimated maximum GPP by 2 to 8 g C m$^2$ d$^{-1}$. (The only exception to this was AU-Lox where the P model predictions that did not account for soil-water stress accurately matched the observed magnitude of GPP; see Supplementary Figure 4. This site is an irrigated orchard.) Model performance also generally improved at semi-arid and even humid sites (Figure 5; Supplementary Figure 8). The RMSE values (Table 1) for arid sites ranged from 0.51 to 1.46 gC m$^2$ d$^{-1}$, compared to 2.07 to 4.01 gC m$^2$ d$^{-1}$ when no stress function was applied. All of the arid sites showed a reduction in RMSE. The RMSE for semi-arid sites ranged from 0.46 to 5.0 gC m$^2$ d$^{-1}$, compared to 1.63 to 5.6 gC m$^2$ d$^{-1}$ when no stress function was applied. All but four of the 22 semi-arid sites showed a reduction in RMSE. The RMSE for humid sites ranged from 1.05 to 5.23 gC m$^2$ d$^{-1}$, compared to 1.75 to 13.08 gC m$^2$ d$^{-1}$ when no stress function was applied. All but five of the 36 humid sites showed a reduction in RMSE.





**Figure 5**. Examples of how the new soil-moisture stress function modifies simulated gross
primary production (GPP$_{new}$) at nine sites representing the range of climatological aridity. The
new model is compared to the simulated level of GPP under well-watered conditions (GPP$_{ww}$),
and to flux-derived values (GPP$_{obs}$). Note that the scale varies between the rows. Plots for all
the flux tower sites are given in Supplementary Figures 6–8.



**Table 1**: Statistics of P model performance (root mean squared error, RMSE) using the new soil moisture stress function (new) and the previous stress function (Pv1.0) from Stocker et al. (2020), compared to P model performance with no soil moisture correction (ww). The sites are grouped by aridity index (AI) classes (see also Supplementary Table 1).

| Site ID | AI | AI class | RMSE (ww) | RMSE (new) | RMSE (v1.0) |
|---|---|---|---|---|---|
| AU-TTE | 7.17 | arid | 2.07 | 0.51 | 0.94 |
| AU-ASM | 6.97 | arid | 2.47 | 0.96 | 1.02 |
| AU-Cpr | 6.36 | arid | 2.83 | 0.77 | 0.87 |
| US-Wkg | 6.34 | not used | 3.93 | 0.9 | 1.86 |
| AU-Lox | 6.32 | not used | 2.15 | 7.03 | 5.79 |
| US-Whs | 5.89 | arid | 3.4 | 0.93 | 1.68 |
| AU-GWW | 5.75 | arid | 2.57 | 0.53 | 1.1 |
| US-SRG | 5.08 | arid | 4.01 | 1.46 | 2.25 |
| US-SRM | 5.02 | arid | 2.82 | 1.04 | 1.45 |
| US-Cop | 3.99 | semiarid | 1.89 | 0.46 | 1.05 |
| AU-Ync | 3.96 | semiarid | 2.75 | 0.67 | 1.7 |
| ES-Ln2 | 3.84 | semiarid | 3.92 | 0.77 | 1.71 |
| AU-Stp | 3.71 | semiarid | 2.62 | 1.33 | 1.44 |
| AU-Emr | 3.08 | semiarid | 4.39 | 1.03 | 2.87 |
| AU-Gin | 2.93 | semiarid | 3.22 | 1.61 | 1.71 |
| AR-SLu | 2.89 | semiarid | 2.07 | 5 | 2.13 |
| ES-LgS | 2.88 | semiarid | 3.33 | 0.78 | 1.69 |
| CN-Du2 | 2.7 | semiarid | 4.53 | 1.47 | 3.02 |
| ZA-Kru | 2.69 | semiarid | 2.14 | 3.3 | 1.82 |
| US-AR2 | 2.61 | semiarid | 3.88 | 1.39 | 2.59 |
| US-AR1 | 2.49 | semiarid | 3.1 | 1.5 | 2.15 |
| AU-Whr | 2.39 | semiarid | 3.13 | 1.41 | 1.63 |
| CN-HaM | 2.34 | semiarid | 1.63 | 1.68 | 1.02 |
| AU-Dry | 2.32 | semiarid | 3.31 | 1.85 | 1.63 |
| IT-Noe | 2.26 | semiarid | 4.04 | 1.61 | 1.86 |
| US-Ton | 2.23 | semiarid | 4.39 | 1.4 | 3.05 |
| US-Var | 2.22 | semiarid | 5.6 | 1.27 | 4.01 |
| ZM-Mon | 2.18 | semiarid | 3.11 | 3.2 | 1.88 |
| AU-RDF | 2.16 | not used | 4.34 | 2.3 | 3.46 |
| US-ARc | 2.04 | semiarid | 3.46 | 2.54 | 2.43 |
| US-ARb | 2.04 | semiarid | 4.02 | 2.91 | 3.05 |
| AU-DaS | 1.81 | humid | 2.3 | 2.9 | 1.56 |
| AU-Rig | 1.81 | humid | 3.91 | 1.81 | 3.45 |
| AU-DaP | 1.8 | humid | 3.76 | 3.21 | 2.66 |




| AU-Wom | 1.75 | humid | 5.63 | 2.26 | 4.25 |
|---|---|---|---|---|---|
| IT-Cp2 | 1.73 | humid | 6.05 | 2.49 | 4.1 |
| AU-Wac | 1.69 | humid | 3.79 | 2.54 | 2.54 |
| FR-Pue | 1.57 | humid | 5.22 | 1.56 | 3.6 |
| AU-Ade | 1.55 | humid | 2.3 | 3.5 | 1.88 |
| AU-How | 1.46 | humid | 2.83 | 3.23 | 2.01 |
| CA-SF3 | 1.41 | humid | 4.38 | 1.12 | 3.61 |
| FR-Fon | 1.39 | humid | 3.04 | 3.39 | 2.59 |
| IT-Col | 1.35 | humid | 4.95 | 3.32 | 3.59 |
| IT-SRo | 1.34 | humid | 4.34 | 2.75 | 2.9 |
| AU-Tum | 1.34 | humid | 4.51 | 3.78 | 3.76 |
| US-KS2 | 1.21 | humid | 13.08 | 5.23 | 12.65 |
| CA-Man | 1.19 | humid | 5.38 | 2.06 | 4.94 |
| CA-NS4 | 1.19 | humid | 4.09 | 1.48 | 3.82 |
| DE-Gri | 1.18 | humid | 2.32 | 2.87 | 2.07 |
| IT-MBo | 1.18 | humid | 4.51 | 2.13 | 4.09 |
| RU-Ha1 | 1.11 | humid | 1.75 | 1.05 | 1.58 |
| FR-LBr | 1.1 | humid | 3.27 | 2.18 | 2.56 |
| US-Wi6 | 1.08 | humid | 5.5 | 2.18 | 5.46 |
| US-PFa | 1.02 | humid | 4.33 | 1.91 | 4.26 |
| AR-Vir | 1.02 | humid | 4.24 | 2.9 | 3.87 |
| US-Syv | 1.01 | humid | 4.88 | 2 | 4.84 |
| RU-Fyo | 0.97 | humid | 2.92 | 2.14 | 2.79 |
| BE-Bra | 0.91 | humid | 3.01 | 1.32 | 3 |
| FI-Hyy | 0.87 | humid | 2.96 | 1.97 | 2.86 |
| NL-Hor | 0.84 | humid | 3.31 | 1.73 | 3.14 |
| CH-Oe1 | 0.8 | humid | 3.67 | 3.94 | 3.67 |
| DE-RuR | 0.78 | humid | 6.42 | 2.96 | 6.4 |
| CZ-BK2 | 0.78 | humid | 5.74 | 3.16 | 5.73 |
| BR-Sa3 | 0.78 | humid | 11.1 | 5.04 | 11.03 |
| BE-Vie | 0.73 | humid | 2.54 | 2.33 | 2.54 |
| CH-Fru | 0.71 | humid | 7.17 | 3.85 | 7.17 |
| IT-Tor | 0.63 | humid | 3.83 | 2.14 | 3.83 |


The new soil-moisture stress function also performed substantially better that the stress
function used in Pv1.0, reducing the overestimation of peak GPP across arid, semi-arid and
humid sites (Figure 6; Supplementary Figures 9-11). The RMSE for arid sites ranged from 0.51
to 1.46 gC m$^2$ d$^{-1}$ compared to 0.87 to 2.25 gC m$^2$ d$^{-1}$ when the Pv1.0 moisture-stress function
was applied. All of these sites showed reduced RMSE. The RMSE for semi-arid sites ranged
from 0.46 to 5.0 gC m$^2$ d$^{-1}$ compared to 1.02 to 4.01 gC m$^2$ d$^{-1}$ when the Pv1.0 moisture-stress



function was applied. All but six of these 22 sites showed reduced RMSE. The RMSE for
humid sites ranged from 1.05 to 5.23 gC m$^2$ d$^{-1}$ compared to 1.56 to 12.65 gC m$^2$ d$^{-1}$ when the
Pv1.0 moisture-stress function was applied. All but eight of these 36 sites showed reduced
RMSE.

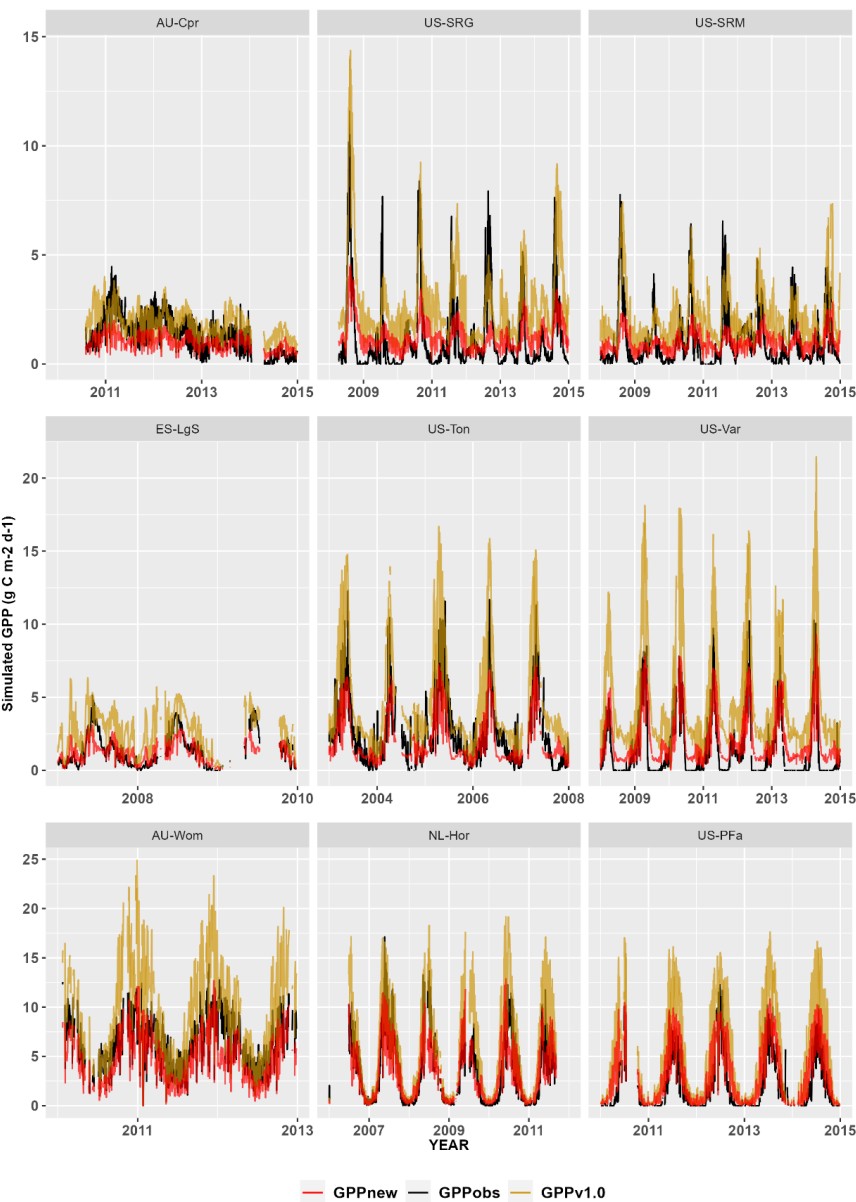

**Figure 6**. Comparison of simulated gross primary production including the new soil-moisture
stress function (GPP$_{new}$) and the original stress function (GPP$_{v1.0}$) from Stocker et al. (2020)
against flux-derived values (GPP$_{obs}$) at nine sites representing the range of climatological
aridity. Note that the scale varies between the rows. Plots for all the flux tower sites are given
in Supplementary Figures 9-11.



**4 Discussion**
We have developed an empirical function to take account of soil-moisture stress in the P model.
The previous introduction of an empirical function to account for soil-moisture stress (Stocker
et al., 2020) produced some improvement in the simulation of GPP by focusing on reducing
GPP when soil moisture was below a critical threshold of the β(θ) ratio. By incorporating a
reduction in the maximum level of the β(θ) ratio with increasing aridity, we have further
improved the performance of the model.
The reduction in the maximum level of LUE with increasing aridity is consistent with the
analyses of Fu et al. (2022), which focused on EF. The climatological aridity index provides a
measure of the degree to which water is likely to be limiting (to both EF and LUE) at some
time during the growing season. The fact that there is a limitation on EF and LUE – even during
intervals with abundant soil moisture – in more arid climates suggests an underlying optimality
principle: that plants adopt water conservation strategies to optimize assimilation over the
whole growing season in the climate to which they are adapted (Manzoni et al., 2011b; Vico
et al., 2013; Fu et al., 2022). Moreover, as also noted by Fu et al. (2022) for EF, the slope of
β(θ) against θ ($y/\psi$ in equation (1)) becomes steeper with increasing aridity. This is a
consequence of the values of the exponent of AI in equations (4) and (5) (0.60 > 0.45, hence
$y/\psi$ is an increasing function of AI). It implies that for every value of AI, there is a value of θ
for which the associated LUE exceeds that of all other β(θ) functions; and that this value
declines as AI increases.
It is well known that some plants continue to photosynthesize at higher levels of drought stress
than others, a behaviour that reflects variability in the strictness of stomatal regulation (Tardieu
and Simmoneau, 1998; McDowell et al., 2008). However, both strict (isohydric) regulation and
less strict (anisohydric) regulation can occur within the same community (e.g. Mediavilla and
Escudero 2003; Cruz de Souza et al., 2020: Raffelsbauer et al., 2023) and species may show
variable regulation over the season and between years (Klein, 2014; Konings and Gentine,
2017). Thus, although there is some evidence that this behaviour is environmentally controlled
(Manzoni et al., 2011; McDowell, 2011; Kumagai and Porporato, 2012; Zhou et al., 2014;
Konings and Gentine, 2017), consistent with our finding that the critical threshold become
lower as climatological aridity increases, it is likely that plant communities often show a
diversity of responses. Our results indicate considerable scatter in both fitted parameters, whose
origin and potential adaptive significance would repay more detailed study.
This work was designed to improve the performance of the P model, which despite its relative
simplicity has been shown to predict the diurnal and seasonal cycles of GPP under well-watered
conditions as well as or better than more complex models (Stocker et al., 2020; Harrison et al.,
2021; Mengoli et al., 2022). How best to represent soil moisture in this context is a challenge.
We have opted for a minimalist approach, using SPLASH. SPLASH is a single-bucket model
that considers only water that is held between the wilting point and field capacity, and does not
account for variation in water holding capacity among soils. The *x*-intercept of the breakpoint
relationship corresponds to the wilting point. We have constrained breakpoint regressions
through the origin since little information was lost by doing so. In reality, the permanent wilting
point varies across species (Koepke et al., 2010; Bartlett et al., 2012) but is also strongly
affected by soil properties (Czyż and Dexter, 2012; Chagas Torres et al., 2021), aspects that
we have ignored. By using a generic soil water-balance model we have also intentionally
decoupled AET (computed by SPLASH on the assumption that the ratio AET/PET is



proportional to relative soil water content) from GPP, thus disregarding the feedback by which
seasonal changes in GPP can influence the seasonal time course of AET and soil moisture. This
research therefore represents a step towards an empirically well-founded representation of the
interactions between carbon and water cycling. A next step will involve the interactive coupling
of transpiration and GPP in a land-surface modelling framework.

*Code and data availability*. The sub-daily P model is implemented in RStudio and is available
on Zenodo (Mengoli G. 2023. https://doi.org/10.5281/zenodo.8018599) and through GitHub
public repository: https://github.com/GiuliaMengoli/P-model_subDaily under the GNU v2.0
license (Mengoli et al. 2022). The new soil moisture stress function and the code to reproduce
the results used in this study is archived on Zenodo (Mengoli G. 2023.
https://doi.org/10.5281/zenodo.8018299) under GNU v2.0 license together with inputs data for
two sites analysed in this study. The code for the SPLASH model v.1.0, in four programming
languages (FORTRAN, C++, Phython, R) is available on Zenodo (Devis et al. 2017.
https://doi.org/10.5281/zenodo.376293) and part of
https://bitbucket.org/labprentice/splash/src/master/ under GNU Lesser General License (Devis
et al. 2017). Meteorological, satellite and gridded climate datasets for this research is available
in these in-text data citation references: Pastorello et al. (2020), [Creative Commons (CC-BY
4.0) license], Stocker B. (2020, December 24), [http://doi.org/10.5281/zenodo.4392703],
Harris et al. (2020), [https://doi.org/10.1038/s41597-020-0453-3]
*Author contributions*. Conceptualization: GM, SPH and ICP; methodology: GM and ICP; data
analysis: GM; writing, first draft: GM and SPH; final draft: all authors.
*Competing interests*. The authors declare no competing interests.
*Financial support and acknowledgments*. GM and ICP acknowledge support from the
European Research Council (787203 REALM) under the European Union's Horizon 2020
research programme. SPH acknowledges support from the ERC-funded project GC2.0 (Global
Change 2.0: Unlocking the past for a clearer future, grant number 694481). This work is a
contribution to the LEMONTREE (Land Ecosystem Models based On New Theory,
obseRvations and ExperimEnts) project, funded through the generosity of Eric and Wendy
Schmidt by recommendation of the Schmidt Futures program. GM acknowledges Carlo Trotta
for providing technical assistance with the code and David Sandoval and Victor Flo for useful
discussions.



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
