# Peer review of "A global function of climatic aridity accounts for soil moisture stress on carbon assimilation"

_EGUsphere, 2023_

## Author Comment (AC1)

Simulations of carbon flux is of high importance but also very difficult. In this study, the authors applied the LUE model to simulate GPP and made modifications on the response of GPP to soil water. This should be interesting to broad readers within Earth science community. But the current depiction in this manuscript did not convince me as their manuscript seems too easy and incomplete. I would suggest a thorough major revision before I can be considered for the target journal.

We are happy that the reviewer also considers that this work should be of broad interest for the modelling community. We believe that the simplicity of our approach is a strength. We will add material to address the reviewer's specific concerns and thus improve the completeness of the manuscript. Below is a point-by-point response to the reviewer's specific comments.

Major points:

1. There is no measure of statistical significance, and the description of the results lacks clarity. Currently, there are statistical numbers (PBIAS, Correlation coefficient), which are very useful.

   A primary aim of our work was to assess quantitatively whether the application of the new function reduced the error in magnitude that commonly affects LUE models (including the P model) when applied in dry environments. RMSE and bias are appropriate statistics for this purpose, as the reviewer recognizes. The improvements of our application are shown to be large. Statistical significance testing is not expected in this context. We therefore included the bias values in the table 1 and revised the caption as follows (the revised table 1 is at the end of this document):

   "Table 1: Statistics of P model performance (root mean squared error, RMSE, and percent bias, PBIAS) using the new soil moisture stress function (new) and the stress function used by Stocker et al. (2020) but applied in the sub-daily model used here, compared to P model performance with no soil moisture correction (ww). The sites are grouped by aridity index (AI) classes (see also Supplementary Table 1). "

   We have also revised the third paragraph of the results to address Point 7 below and to clarify the basis for the evaluation of the results, and the caption of Fig. 2.

   " Figure 2. Box-plot comparison of the fitted…Letters indicate whether the median values are significantly different based on the Kruskal-Wallis test, $P < 0.05$. Classes that are significantly different from one another are indicated by different letters."

   We have added a sentence, to clarify the $p$-value threshold we used for the non-parametric test:

   "The non-parametric Kruskal-Wallis test was used to determine whether there were significant differences in fitted parameter values among aridity classes. We used a $P < 0.05$ to identify significant differences between the aridity groups."

2. In the methods part, the description of the P model is incomplete. In addition, the authos should add the calculation of GPP in the model and describe the design of expriment and model setting in detail.

The P model has been fully described in other publications: the original version in GMD (Stocker et al., 2020) and the sub-daily version, used here, in JAMES (Mengoli et al., 2022). In order to avoid repeating information that has already been published, we only provide a general description of the P model in the Methods section of this paper to describe the key characteristics of the model. However, this comment made us realize that it would be useful to clarify one important aspect of this work, which is that the approach we have taken is not simply a modification of the P model but has more general utility. We have modified the abstract to make this clear as follows:

"….Substantially improved GPP simulation is shown during both unstressed and water-stressed conditions, compared to the reference model version that ignores soil-moisture stress, and to an earlier formulation in which maximum LUE was not reduced. Our results demonstrate (a) how climatic aridity modulates the response of GPP to soil moisture independently of plant functional types and (b) that this modulation satisfies an optimality criterion: i.e. that for any aridity value there is a soil moisture value at which the associated GPP response is maximal. These lessons are transferable to any LUE-based model, with re-calibration of the functions as required. These results point the way towards a better approach to the simulation of soil moisture stress in different models and a better-founded representation of carbon-water cycle coupling in any vegetation or land-surface model."

And we have also included some additional explanatory text at the end of the introduction:

"….The performance of the resulting model is compared with that of the uncorrected P model, and with a version of the sub-daily model that applies the soil-moisture stress function previously developed by Stocker et al. (2020). Results show that the function relating the reduction of assimilation due to low soil moisture varies systematically as a function of climatic aridity rather than being dependent on the type of vegetation. Moreover, the GPP reduction satisfies an optimality criterion: i.e. that for any aridity values there is a soil moisture value at which the associated GPP response function is maximal. The new function provides a promising approach to including soil moisture effects in a modelling framework and could be applied in other vegetation and land-surface models."

We have also revised the discussion as follows:

"This work was originally designed to improve the performance of the P model, which despite its relative simplicity has been shown to predict the diurnal and seasonal cycles of GPP under well-watered conditions as well as or better than more complex models (Stocker et al., 2020; Harrison et al., 2021; Mengoli et al., 2022). However, our intent is to present an approach that relies on a simple algorithm that could have more general utility in a modelling context."

We have also written the conclusion section, in response to comment 9 (see below), to highlight the general utility of the approach we are presenting.

We provided the important details about the design of experiment and model setting in the original version of the manuscript. However, to make it clear that we are using a sub-daily version of the P model, we have specifically named this version and we have expanded the description in the methods section, as follows:

"….The P model (Stocker et al., 2020) was modified by Mengoli et al. (2022) in order to simulate diurnal cycles, separating the instantaneous responses of GPP (with photosynthetic parameters fixed over the diurnal cycle) from the acclimation responses of those parameters on a time scale of around two weeks. This modified model (P-model subDaily v1.0.0) is used here to simulate daily GPP, as the daily sum of GPP computed on half-hourly timesteps. The sub-daily model can be run in two modes, either by using an exponential-weighted mean of the acclimating quantities or by using a 15-day running mean of midday temperature to determine acclimation. The two methods produce virtually identical results (Mengoli et al., 2022). Here we use a 15-day running mean of midday temperature to determine acclimation. Mengoli et al (2022) showed that the P-model subDaily v1.0.0 accurately reproduces the diurnal cycle of GPP in well-watered sites but tends to overestimate GPP in drylands, because it does not account for any soil moisture limitation on GPP."

Then we have revised the text of the MS to introduce the version name of the sub-daily model (P-model subDaily v1.0.0) we used for the analysis. We have also re-ordered 2.1 section in the methods to clarify the difference between this version of the model and the original version of the model, and particularly with respect to the soil moisture stress function used in the original model, as follows:

"The P model is a LUE model based on eco-evolutionary optimality theory for the trade-off between carbon uptake and water loss (Prentice et al., 2014) and the acclimation and/or adaptation of leaf-level photosynthesis to environmental conditions (Wang et al., 2017). The model is driven by air temperature, vapour pressure deficit (VPD), incident photosynthetic photon flux density (PPFD), the fraction of incident PPFD absorbed by leaves (fAPAR), elevation (to calculate atmospheric pressure) and the ambient partial pressure of carbon dioxide (ca). The model distinguishes $C_3$ and $C_4$ photosynthesis but does not require specification of distinct parameter values of any other plant functional types. When driven by satellite-derived fAPAR, it reproduces the seasonal cycle and interannual variability in GPP at flux sites from a range of natural vegetation types as well as geographic variation in GPP (Wang et al., 2014; Balzarolo et al., 2019; Stocker et al., 2020) and temporal trends in GPP at flux sites (Cai & Prentice, 2020).

The P model (Stocker et al., 2020) was modified by Mengoli et al. (2022) in order to simulate diurnal cycles, separating the instantaneous responses of GPP (with photosynthetic parameters fixed over the diurnal cycle) from the acclimation responses of those parameters on a time scale of around two weeks. This modified model (P-model subDaily v1.0.0) is used here to simulate daily GPP, as the daily sum of GPP computed on half-hourly timesteps. The sub-daily model can be run in two modes, either by using an exponential-weighted mean of the acclimating quantities or by using a 15-day running mean of midday temperature to determine acclimation. The two methods produce virtually identical results (Mengoli et al., 2022). Here we use a 15-day running mean of midday temperature to determine acclimation. Mengoli et al (2022) showed that the P-model subDaily v1.0.0 accurately reproduces the diurnal cycle of GPP in well-watered sites but tends to overestimate GPP in drylands, because it does not account for any soil moisture limitation on GPP.

Given the known tendency of the P model to overestimate GPP under dry conditions, the FULL configuration of the current standard P model Pv1.0 (Stocker et al., 2020) includes an empirical water stress function (also based on eddy-covariance flux data) that approaches 1 at a threshold value of θ (θ*), where θ is plant-available water expressed as a fraction of soil water-holding capacity, and θ* is set to 0.6. The function declines more steeply with decreasing θ in drier climates, with climatic moisture quantified by an estimate of the ratio (α) of actual evapotranspiration (AET) to potential evapotranspiration (PET). This function is used in Pv1.0 (FULL) as a multiplier of the modelled, well-watered GPP, in a similar way to the function proposed here, but has not been applied in the sub-daily model used here."

And we have emphasized where the model settings can be found rephrasing the sentence under the code and data availability section as follows:

"….The code for the new soil moisture stress function and for running the P-model subDaily v1.0.0 version as applied in this study is archived on Zenodo…"

3. The organization of the manuscript and presentation of the results need some improvement, current manuscript seems too easy, and the authos can add another part of results about regional comparison between the original model, the revised model and remote sensing GPP products.

We included comparisons between the original sub-daily model, the sub-daily model including the new stress function and the sub-daily model using the original stress function from Stocker et al. (2020) in the original manuscript. However, we agree that comparisons with another remote sensing GPP product would be useful and have now added a comparison with the current, improved version of the widely used MODIS GPP product. We show that our model compares very well with the MODIS product, but has the advantage of being theory based and parameter sparse.

We have revised the abstract to indicate that we have made this comparison:

"… Substantially improved GPP simulation is shown during both unstressed and water-stressed conditions, compared to the reference model version that ignores soil-moisture stress, and to an earlier formulation in which maximum LUE was not reduced. The model performance is similar to that of the most recent version of the MODIS GPP product."

We have modified the introduction to indicate that we have made this comparison:

"…The performance of the resulting model is compared with that of the uncorrected P model, with a version of the sub-daily model that applies the soil-moisture stress function previously developed by Stocker et al. (2020) and with the MODIS remotely sensed GPP product (Running and Zhao, 2021)."

We have modified the section 2.7 of the methods section as follows:

"…We compared the predictions of GPP obtained using this new soil-moisture stress function to GPP simulated by the P-model subDaily v1.0.0 with (a) no soil-moisture stress and (b), the soil-moisture stress function used in Pv1.0, at all of the flux-tower sites, with meteorological data provided for the site in the FLUXNET2015 data set and fAPAR data from Stocker et al.

(2020). We also compared these predictions with the current, improved version of the widely used MODIS GPP product (MOD17A2HGF v0.61: Running and Zhao, 2021; https://doi.org/10.5067/MODIS/MOD17A2HGF.061) The goodness-of-fit between each of the modelled estimates of GPP and the flux-derived GPP at each site was quantified by the root mean squared error (RMSE)."

We have included an additional paragraph to describe the comparison with the MODIS product in the results, including a new figure (Fig.7) and table (Table 2) in the main text and additional figures in the SI, as follows:

" Comparison of the new soil-moisture stress function with MODIS GPP shows a similar level of performance (Figure 7; Supplementary Figures 12-14). The average RMSE for the P model and MODIS at arid sites is 6.67 and 5.80 gC $m^2$ $8$-$d^{-1}$ respectively, and the range of RMSE values (Table 2) is comparable (3.51-11.08 gC $m^2$ $8$-$d^{-1}$ for the P model; 3.50-9.83 gC $m^2$ $8$-$d^{-1}$ for MODIS GPP). The average RMSE at semi-arid sites is 13.98 and 14.66 gC $m^2$ $8$-$d^{-1}$ for the P model and MODIS respectively, with ranges between 3.84-39.41 gC $m^2$ $8$-$d^{-1}$(P model) and 4.62-51.60 gC $m^2$ $d^{-1}$ (MODIS GPP). The average RMSE at humid sites is 19.61 and 16.80 gC $m^2$ $8$-$d^{-1}$ for the P model and MODIS, respectively, and again the ranges are comparable (P model: 6.17 to 40.95 gC $m^2$ $8$-$d^{-1}$ ; MODIS 6.40 to 30.06 gC $m^2$ $8$-$d^{-1}$). "

We have also added text in the discussion:

"We have developed an empirical function to take account of soil-moisture stress in the P model. The previous introduction of an empirical function to account for soil-moisture stress (Stocker et al., 2020) produced some improvement in the simulation of GPP by focusing on reducing GPP when soil moisture was below a critical threshold of the β(θ) ratio. By incorporating a reduction in the maximum level of the β(θ) ratio with increasing aridity, we have further improved the performance of the model when using this function in the sub-daily model. The performance of the P-model subDaily v1.0.0 is similar to that of the most recent and improved gap-filled version of MODIS GPP (MOD17A2HGF v 0.61). MODIS is a widely used product but uses a PFT-specific parametrization, whereas the P model makes no distinctions by PFTs. Furthermore, whereas MODIS is empirically based, the P model has a strong theoretical basis in eco-evolutionary optimality theory, allowing it to take account of the impact of changing $CO_2$ on assimilation in a natural way. Thus, our theory-based and parameter-sparse model provides an alternative approach that performs as well as the MODIS product. "

We have included the source of the MODIS data in the code and data availability statement:

"Meteorological, satellite and gridded climate datasets for this research is available in these in-text data citation references: Pastorello et al. (2020), [Creative Commons (CC-BY 4.0) license], Stocker B. (2020, December 24), [http://doi.org/10.5281/zenodo.4392703], Harris et al. (2020), [https://doi.org/10.1038/s41597-020-0453-3], Running & Zhao (2021) , [https://doi.org/10.5067/MODIS/MOD17A2HGF.061]"

We added the appropriate citation to the reference list:

"Running, S., Zhao, M.: MODIS/Terra Gross Primary Productivity Gap-Filled 8-Day L4 Global 500m SIN Grid V061 [Data set]. NASA EOSDIS Land Processes Distributed Active

Archive Center. Accessed 2023-09-28 from https://doi.org/10.5067/MODIS/MOD17A2HGF.061, 2021 "

The caption of the new figure 7 is:

"**Figure 7.** Comparison of simulated gross primary production including the new soil-moisture stress function (GPP$_{new}$) and the gross primary production simulated by MOD17A2HGF v0.61 (GPP$_{MODIS}$) against flux-derived values (GPP$_{obs}$) at nine sites representing the range of climatological aridity. Note that the scale varies between the rows. Plots for all the flux tower sites are given in Supplementary Figures 12-14."

The captions of the new SI figures (note that the figures are displayed at the end of this document together with the new table 2) are:

Supplementary Figure 12: Comparison of simulated gross primary production including the new soil-moisture stress function (GPPnew) and the gross primary production simulated by MOD17A2HGF v0.61 (GPP$_{MODIS}$) against flux-derived values (GPPobs) at flux tower sites classified as arid (aridity index, AI > 5).

Supplementary Figure 13: Comparison of simulated gross primary production including the new soil-moisture stress function (GPPnew) and the gross primary production simulated by MOD17A2HGF v0.61 (GPP$_{MODIS}$) against flux-derived values (GPPobs) at flux tower sites classified as semi-arid (aridity index, AI = between 2 and 5).

Supplementary Figure 14: Comparison of simulated gross primary production including the new soil-moisture stress function (GPPnew) and the gross primary production simulated by MOD17A2HGF v0.61 (GPP$_{MODIS}$) against flux-derived values (GPPobs) at flux tower sites classified as humid (aridity index, AI < 2).

[Figure]

4. In the abstract, the authors didn't fully explain the performances of the revised model, and the abstract looks unat [*sic*]

We have re-written part of the abstract in order to emphasize the performance of the revised model and also to highlight the more general implications of this research. Please see text in the response to the second comment.

5. Introduction: Line 71-88: Poorly literature citation, relevant literature on critical drought thresholds that affect GPP should be added. (Li et al., 2023. Global variations in critical drought thresholds that impact vegetation)

We recognize that our presentation of the recent literature on critical drought thresholds was rather brief. However, much of this literature focuses on greenness whereas our focus is on quantifying the effect of soil moisture on the light-use efficiency of GPP. In fact, the P model uses greenness as input. We have clarified this in the new paragraph in the introduction, and expanded our general discussion of the literature as follows:

"There is evidence that soil moisture, rather than atmospheric demand, is the principal constraint on GPP in arid and semi-arid ecosystems (Xu et al., 2023; Pei et al., 2020; Dubey & Ghosh 2023). It has also been shown that GPP is substantially reduced (much more than total ecosystem respiration) in response to drought (e.g. Shi et al., 2020). Liu et al. (2020) showed that soil moisture is the dominant water stress on vegetation over 70% of the global land area. However, the response of GPP to water stress in models from the previous round of the Coupled Model Intercomparison Project, CMIP5, is too strong (Huang et al., 2016) and representation of the soil moisture effects on GPP remains one of the largest sources of uncertainty in carbon cycle models (Trugman et al., 2018). Many studies have focused on the impact of drought on vegetation greenness (e.g. Li et al., 2023), but soil moisture stress also impacts light-use efficiency (LUE) directly, which further reduces GPP (Lv et al., 2023; Xing et al., 2023). Thus, it is important to take account of the impact of soil moisture stress on LUE as well as on vegetation greenness."

We have updated the reference list including these new references.

6. Figure1: Although the authors use breakpoint regression analysis, I hardly find the relationship between scatters and fitting lines.

Breakpoint regression analysis has been widely used for this purpose, i.e. to identify critical soil-moisture thresholds below which there is a steep decline of evaporative fraction and/or GPP. We performed breakpoint regression analysis based on the data for each site independently, so these "broken-stick'' models are the actual fitted regressions based on the data as shown in Figure 1. The technique is powerful, able to extract information from noisy data that would be impossible to derive by visual inspection. The results are coherent internally and with other published results – showing that plants' ability to extract water from dry soils, supporting photosynthesis, is enhanced as climatic aridity increases. However, we agree that it was hard to

distinguish where the bulk of the data was concentrated in the original figures and that reader could be confused by the scatter in the observations. We have produced a new version of the figure where we use a heat map approach to show the density of the data points - this makes it clearer that the break point analysis identifies threshold based on the region where there is a high density of observations. The new figure is given in the response to the second reviewer.

7. Line 265-278: The description in this paragraph makes it difficult to see the improvement of the revised model compared to the original model, and the presentation must be revised. Currently, there are statistical numbers(PBIAS, Correlation coefficient), which are very useful. In addition, the average percentage reduction in RMSE in arid, semi-arid, and humid regions should be calculated separately for further clarification.

We have revised the paragraph to further clarify the text generally as noted above, and we have also provided the RMSE reduction for the three aridity classes as suggested. The new text reads as follows:

" Implementation of the new empirical soil-moisture stress function produced a substantial improvement in model performance compared to simulations with no soil-moisture stress function (Figure 6, Supplementary Figures 6-8). At sites classified as arid (AI >5), simulations that did not account for soil-water stress produced an overestimation of maximum GPP between 2 and 8 gC m$^2$ d$^{-1}$. (The only exception to this was AU-Lox where the P model predictions that did not account for soil-water stress accurately matched the observed magnitude of GPP; see Supplementary Figure 4. This site is an irrigated orchard). The overestimation of peak GPP at sites classified as semi-arid (AI between 2-5) was of a similar magnitude (2 to 10 gC m$^2$ d$^{-1}$). Even at sites classified as humid (AI < 2), there was a noticeable improvement in performance at most sites (Figure 6; Supplementary Figure 8). The improved performance compared to the version of the P model with no soil-moisture stress function is reflected in the RMSE values (Table 1). The RMSE for arid sites ranged from 0.51 to 1.46 gC m$^2$ d$^{-1}$ compared to 2.07 to 4.01 gC m$^2$ d$^{-1}$ when no stress function was applied. All of the arid sites showed a reduction in RMSE, with an average reduction in RMSE of 69.26%. The RMSE for semi-arid sites ranged from 0.46 to 5.0 gC m$^2$ d$^{-1}$ compared to 1.63 to 5.6 gC m$^2$ d$^{-1}$ when no stress function was applied. All but four of the 21 semi-arid sites showed a reduction in RMSE, with an average reduction in RMSE of 47.28%. The RMSE for humid sites ranged from 1.05 to 5.23 gC m$^2$ d$^{-1}$ compared to 1.75 to 13.08 gC m$^2$ d$^{-1}$ when no stress function was applied. All but five of the 36 humid sites showed a reduction in RMSE, with an average reduction of 42.1%.

8. I admit the revised model reduced the overestimation of GPP compared to the original model, however, the revised model can't capture the peaks of GPP, and at some sites (US-SRG, US-Var), the revised model overestimated GPP in the non-growing season. The authors should be further analyzed these results.

It is true that the model does not properly capture some peak values of GPP and overestimates GPP outside the growing season at some sites. We will raise these issues in the Discussion as follows:

*"The application of the new function substantially reduces the overestimation of GPP compared to the original model, and to the moisture stress function developed by Stocker et al. (2020) when applied in the sub-daily model. However, the model does not always capture peaks in GPP shown by the observations; it also overestimates GPP outside the growing season at some sites (e.g. US-Var). It is difficult to identify the causes of specific mismatches between eddy-covariance-derived and simulated GPP on particular days or weeks because such mismatches can have multiple causes. In addition to possible issues with the model itself, there is uncertainty in the partitioning of measured net ecosystem exchange to GPP versus ecosystem respiration (particularly during the non-growing season) and unavoidable discrepancies between the satellite-derived pixel data and the footprint of the flux tower."*

9. The conclusion part should be added to the manuscript.

We will add a Conclusion section as follows:

"We have derived a new empirical function to account for the soil moisture effect on the light use efficiency of GPP as a function of climatological aridity. The new function provides a constraint on both the maximum level of GPP and on the critical soil moisture threshold with increasing climatological aridity. Climatological aridity provides a measure of the degree to which water is likely to be limiting at some time during the growing season. The new formulation is thus consistent with the idea that plants adopt water conservation strategies to optimize assimilation over the whole growing season in the climate to which they are adapted. The new formulation produces improved simulation of GPP at flux tower site from arid, semiarid and humid regions, both during water-stressed conditions and during unstressed periods. Although this new function is tested in the context of the existing LUE model (the P model), it is generic and could easily be applied in other models, including land-surface schemes."

New Figures and Tables

Supplementary Figure 12: Comparison of simulated gross primary production including the new soil-moisture stress function (GPPnew) and the gross primary production simulated by MOD17A2HGF v0.61 (GPP$_{MODIS}$) against flux-derived values (GPPobs) at flux tower sites classified as arid (aridity index, AI > 5).

Supplementary Figure 13: Comparison of simulated gross primary production including the new soil-moisture stress function (GPPnew) and the gross primary production simulated by MOD17A2HGF v0.61 (GPP$_{MODIS}$) against flux-derived values (GPPobs) at flux tower sites classified as semi-arid (aridity index, AI = between 2 and 5).

Supplementary Figure 14: Comparison of simulated gross primary production including the new soil-moisture stress function (GPPnew) and the gross primary production simulated by MOD17A2HGF v0.61 (GPP$_{MODIS}$) against flux-derived values (GPPobs) at flux tower sites classified as humid (aridity index, AI < 2).

Table 2: Statistics of P model performance (root mean squared error, RMSE, and percent bias, PBIAS) using the new soil moisture stress function (new) compared to MOD17A2HGF v0.61performance (MODIS). The sites are grouped by aridity index (AI) classes (see also Supplementary Table 1).

[Figure]

[Figure]

[Figure]

[Figure]

| Site ID | AI | ARIDITY | RMSE(new) | RMSE(MODIS) | PBIAS (new) | PBIAS (MODIS) |
|---|---|---|---|---|---|---|
| AU-TTE | 7.17 | arid | 4.01 | 4.27 | 702.7 | 692.2 |
| AU-ASM | 6.97 | arid | 7.51 | 5.96 | -2.2 | 20.7 |
| AU-Cpr | 6.36 | arid | 5.51 | 4.45 | -24.7 | 18.3 |
| US-Wkg | 6.34 | not used | 6.96 | 6.78 | 14.2 | -19.4 |
| AU-Lox | 6.32 | not used | 52.43 | 41.96 | -75.5 | -57 |
| US-Whs | 5.89 | arid | 7.19 | 5.49 | 74.6 | 32.7 |
| AU-GWW | 5.75 | arid | 3.51 | 3.45 | -18.1 | 7.4 |
| US-SRG | 5.08 | arid | 11.08 | 9.83 | 7.4 | -7.1 |
| US-SRM | 5.02 | arid | 7.86 | 7.13 | -5.4 | -4.6 |
| US-Cop | 3.99 | semiarid | 3.84 | 4.62 | 112.2 | 145.2 |
| AU-Ync | 3.96 | semiarid | 7.99 | 9.91 | 156.1 | 241.9 |
| ES-Ln2 | 3.84 | semiarid | 6.86 | 9.92 | 1080.4 | 1594.5 |
| AU-Stp | 3.71 | semiarid | 10.11 | 7.62 | -15.4 | 19.8 |
| AU-Emr | 3.08 | semiarid | 7.75 | 11.1 | 47.6 | 84.8 |
| AU-Gin | 2.93 | semiarid | 12.56 | 8.6 | -40.8 | 18.1 |
| AR-SLu | 2.89 | semiarid | 39.41 | 51.59 | -56.3 | -74.2 |
| ES-LgS | 2.88 | semiarid | 5.49 | 8.27 | -10.7 | 47.7 |
| CN-Du2 | 2.7 | semiarid | 11.15 | 9.28 | 89.8 | 63.8 |
| ZA-Kru | 2.69 | semiarid | 23.02 | 15.19 | -51.3 | 12 |
| US-AR2 | 2.61 | semiarid | 10.04 | 9.05 | 61.2 | 15.2 |
| US-AR1 | 2.49 | semiarid | 15.48 | 15.85 | -17.4 | -29.1 |
| AU-Whr | 2.39 | semiarid | 10.64 | 6.86 | -35.3 | -9.2 |
| CN-HaM | 2.34 | semiarid | 12.64 | 14.71 | -40.8 | -56.8 |
| AU-Dry | 2.32 | semiarid | 15.49 | 12.06 | -40.5 | -17.6 |
| IT-Noe | 2.26 | semiarid | 12.55 | 19.84 | -39.7 | 49.7 |
| US-Ton | 2.23 | semiarid | 10.02 | 8.71 | -21 | 5.1 |
| US-Var | 2.22 | semiarid | 9.52 | 13.97 | 39.2 | 64.4 |
| ZM-Mon | 2.18 | semiarid | 24.53 | 17.68 | -50.2 | -14.7 |
| AU-RDF | 2.16 | not used | 17.93 | 23.67 | -1.6 | 28.7 |
| US-ARb | 2.04 | semiarid | 25.27 | 29.53 | -21.4 | -34 |
| US-ARc | 2.04 | semiarid | 19.62 | 23.62 | -22.9 | -32.8 |

| | | | | | |
|---|---|---|---|---|---|
| AU-DaS | 1.81 | humid | 21.7 | 22.01 | -48.1 | -31.3 |
| AU-Rig | 1.81 | humid | 13.51 | 14.97 | -6.2 | -2.3 |
| AU-DaP | 1.8 | humid | 24.87 | 23.36 | -32.1 | -6.4 |
| AU-Wom | 1.75 | humid | 15.75 | 14.04 | -23.7 | 4.7 |
| IT-Cp2 | 1.73 | humid | 22.62 | 14.21 | -33.5 | -2.5 |
| AU-Wac | 1.69 | humid | 19.49 | 20.18 | -37.3 | 19 |
| FR-Pue | 1.57 | humid | 11.57 | 10.92 | -14.6 | 8.2 |
| AU-Ade | 1.55 | humid | 26.83 | 24.76 | -52.8 | -43 |
| AU-How | 1.46 | humid | 24.62 | 17.08 | -51.2 | -22 |
| CA-SF3 | 1.41 | humid | 6.17 | 12.08 | 8.2 | 23.8 |
| FR-Fon | 1.39 | humid | 26.27 | 12.05 | -34.2 | 3.6 |
| IT-Col | 1.35 | humid | 25.12 | 20.47 | -23.9 | 0.9 |
| AU-Tum | 1.34 | humid | 28.98 | 22.33 | -32.4 | -16.3 |
| IT-SRo | 1.34 | humid | 20.78 | 14.64 | -34.7 | -13.7 |
| US-KS2 | 1.21 | humid | 40.95 | 17.2 | 88.1 | 24.6 |
| CA-Man | 1.19 | humid | 21.68 | 16.62 | 85.6 | 33.5 |
| CA-NS4 | 1.19 | humid | 10.51 | 8.85 | 45.1 | 29.7 |
| DE-Gri | 1.18 | humid | 21.36 | 18.19 | -34.9 | -27.8 |
| IT-MBo | 1.18 | humid | 15.58 | 13.19 | -3.1 | -18.4 |
| RU-Ha1 | 1.11 | humid | 7.48 | 6.4 | -16.4 | -14.6 |
| FR-LBr | 1.1 | humid | 15.57 | 12.29 | -30.6 | -20.2 |
| US-Wi6 | 1.08 | humid | 14.74 | 21.4 | 59.1 | 86.6 |
| AR-Vir | 1.02 | humid | 23.03 | 26.32 | -21.5 | -20 |
| US-PFa | 1.02 | humid | 13.32 | 16.33 | 46.6 | 54.7 |
| US-Syv | 1.01 | humid | 13.98 | 10.97 | 15.6 | -9.7 |
| RU-Fyo | 0.97 | humid | 15.41 | 15.45 | -20.1 | -26.6 |
| BE-Bra | 0.91 | humid | 9.3 | 6.62 | -6.8 | 0 |
| FI-Hyy | 0.87 | humid | 14.8 | 10.17 | -11.7 | -18.9 |
| NL-Hor | 0.84 | humid | 13.34 | 12.57 | -6.1 | -6.8 |
| CH-Oe1 | 0.8 | humid | 28.33 | 30.06 | -30.2 | -37.2 |
| BR-Sa3 | 0.78 | humid | 38.47 | 26.22 | 31 | -21 |
| CZ-BK2 | 0.78 | humid | 20.05 | 19.45 | 24.1 | 21.3 |
| DE-RuR | 0.78 | humid | 21.85 | 16.78 | 37.1 | -24.8 |

| | | | | | |
|---|---|---|---|---|---|
| BE-Vie | 0.73 | humid | 16.21 | 19.69 | -22.5 | -35.2 |
| CH-Fru | 0.71 | humid | 28.03 | 23.37 | 44.5 | -34.8 |
| IT-Tor | 0.63 | humid | 13.73 | 13.66 | 38.6 | -34.7 |

---

## Author Comment (AC2)

This development and technical paper concerns the development of a new model to simulate GPP which does not directly use soil moisture as limit to carbon assimilation, but rather uses aridity to account for soil moisture stress on carbon assimilation.

The reviewer has misunderstood our method. We do indeed use soil moisture directly as a limitation on carbon assimilation. However, the function relating the degree of assimilation reduction due to low soil moisture is empirically shown to vary systematically as a function of climatic aridity. As this is a fundamental point, we have tried to make this logic clear already in the revised Abstract:

"The coupling between carbon uptake and water loss through stomata implies that gross primary production (GPP) can be limited by soil water availability through reduced leaf area and/or reduced stomatal conductance. Vegetation and land-surface models typically assume that GPP is highest under well-watered conditions and apply a stress function to reduce GPP with declining soil moisture below a critical threshold, which may be universal or prescribed by vegetation type. It is unclear how well current schemes represent the water conservation strategies of plants in different climates. Here eddy-covariance flux data are used to investigate empirically how soil moisture influences the light-use efficiency (LUE) of GPP. Well-watered GPP is estimated using a first-principles LUE model driven by atmospheric data and remotely sensed green vegetation cover, the P model. Breakpoint regression is used to relate the daily value of the ratio $\beta(\theta)$ (flux-derived GPP/modelled well-watered GPP) to soil moisture, which is estimated using a generic water-balance model. Although the soil moisture is used directly as a limitation on carbon assimilation the function relating the degree of assimilation reduction due to low soil moisture is empirically shown to vary systematically as a function of climatic aridity. Maximum LUE, even during wetter periods, is shown to decline with increasing climatic aridity index (AI). The critical soil-moisture threshold also declines with AI. Moreover, for any AI, there is a value of soil moisture at which $\beta(\theta)$ is maximized, and this value declines with increasing AI. Thus, ecosystems adapted to seasonally dry conditions use water more conservatively (relative to well-watered ecosystems) when soil moisture is high, but maintain higher GPP when soil moisture is low. An empirical non-linear function of AI expressing these relationships is derived by non-linear regression, and used to generate a $\beta(\theta)$ function that provides a multiplier for well-watered GPP as simulated by the P model. Substantially improved GPP simulation is shown during both unstressed and water-stressed conditions, compared to the reference model version that ignores soil-moisture stress, and to an earlier formulation in which maximum LUE was not reduced… Our results demonstrate (a) how climatic aridity modulates the response of GPP to soil moisture independently of plant functional types and (b) that this modulation satisfies an optimality criterion: i.e. that for any aridity value there is a soil moisture value at which the associated GPP response is maximal. These lessons are transferable to any LUE-based model, with re-calibration of the functions as required. These results point the way towards a better approach to the simulation of soil moisture stress in different models and a better-founded representations of carbon-water cycle coupling in any vegetation or land-surface model."

And we added the following sentence in the last paragraphs of the introduction:

"…These relationships are used to generate a family of $\beta(\theta)$ functions, dependent on AI, which can serve as multipliers of the modelled, well-watered GPP. We used the P model to simulate GPP, and

soil moisture directly as limitation on carbon assimilation, when we apply the new empirical function. The performance of the resulting…"

We have also modified the text in section 2.4 (Breakpoint regression analysis):

"….
$$\beta(\theta) \ = \ \min \ [y, \ (y/\psi) \times \theta] \tag{1}$$

where $\beta(\theta)$ is equal to its maximum level ($y$) when $\theta \geq \psi$ while it is equal to the ratio between its maximum level and the critical threshold ($y/\psi$) when $\theta < \psi$. "

Major comments:

- Following GMD paper conventions, a development and technical paper should be clear on which model/code it is improving. I assume in this case it is the "P model"? That should be emphasized in the title. It also lacks details on the "technical aspects of running models and the reproducibility of results", or "a significant amount of evaluation against standard benchmarks, observations, and/or other model output", see https://www.geoscientific-model-development.net/about/manuscript_types.html#item2. Discussing how the code can be used is a feature distinct to GMD papers, which is not provided in this preprint.

  A key point here is that our results have relevance beyond the P model; they point the way towards a better approach to the simulation of soil moisture effects in different models, including LUE models, land-surface schemes and DGVMs. Therefore, it does not seem appropriate that the title should include the specific version name. However, we have included the version name in the revised subsections of the method part of the MS and, importantly, we have revised the text to make the wider relevance of the work more apparent. The amended text in the abstract, introduction, and discussion on the general utility of this approach is given in the response to the reviewer 1 (point 2). We have also commented on the general utility of this approach in the new conclusion, provided in response to comment 9 of reviewer 1.

  We believe that the structure of the Methods allows readers to understand the steps required to reproduce our work. However, we have added a new final paragraph in the Methods, in response to point 2 raised by the reviewer 1, which provides information on the settings used to run the model.

  Regarding how the code should be used, we followed the journal guidelines and included a section (*Code and data availability*) in the original manuscript to document the different sources of code and data. Specific details are given in the readme files on the Zenodo/GitHub repositories. The revised text for this section is given in the response to reviewer 1.

The original manuscript included evaluation against observations, and other models. In response to a comment by the first reviewer, we have now added a comparison of our model with the latest version of MODIS GPP. Please see response to reviewer 1 (comment 3) for the added figures and text supporting this new evaluation.

- Code availability: it is not entirely clear whether the provided code is an improved version of the "P model" or a standalone module that can be attached to the "P model".

  We used the sub-daily version of the P model, which is an enhanced version of the P model v1.0 (Stocker et al. 2020, GMD) adapted to work at a sub-daily timescale. The sub-daily version is a stand-alone model that only needs meteorological and satellite input data to simulate GPP. The new function for the soil moisture response, presented here, is an external module that can be applied to the subdaily model output directly. This new module, and its application, have been deposited and are publicly available in the GitHub channel cited in the MS. There are readme files for all the codes provided. Specific modifications to the text which clarify which version of the model we use and how it is run have been made in response to comment 2 by reviewer 1.

- An impression from reading this manuscript is that it reads like an report, rather than an academic paper. It only discusses work that are immediately related to the approach taken, but hasn't provided a survey of the full suite of other work that has contributed to the field.
  In response to a similar comment by reviewer 1 (comment 5), we have expanded the introduction to provide a fuller coverage of other work in this area.

- The sensitivity of the model parameters should be investigated, especially since the new model basically relies on a single scaling parameter beta. Also, a potential issue with using aridity index is that it is an long-term average, which may not capture changing aridity under changing climate.

  The first point, as we understand it, addresses the issue of uncertainty in the values of $\beta(\theta)$ derived using our empirical functions. In other words: to what extent does this uncertainty influence the resulting simulations of GPP? We therefore performed a simple sensitivity test to show the order of magnitude of this effect in arid versus humid climates. We describe the results of this sensitivity analysis in the results section as follows:

  "We performed a sensitivity test to assess the impact of uncertainty in the estimated parameters on GPP, by substituting the upper and lower value of the standard errors on the fitted parameters in equation (4) and (5). This test showed that these uncertainties had little impact on $\beta(\theta)$ and did not change the simulated GPP (Figure 5)."

  The caption for the new figure (displayed at the end of this comment) is:

Figure 5: Sensitivity of the model to parameter uncertainty. The plot shows gross primary production (GPP) using the new soil-moisture stress function (GPPnew) at six sites representing the range of climatological aridity compared to the simulated GPP resulting from the adding the upper (GPPnew +) and lower (GPPnew -) standard error to the canonical fitted parameters in equation 4 and 5. The flux-derived values (GPPobs) are also shown. Note that the scale varies between the rows.

The second point is also a good one. Aridity index (AI) is a long-term average. Its use here reflects the finding (see also the cited papers by Fu et al.) that the "fast" soil-moisture response of plant function (whether evaporative fraction or GPP) differs between climates – indicating that the vegetation in climates differing in aridity is differentially adapted to cope with low soil moisture. (This is not a controversial idea in itself, but current land surface models capture it – if at all – only through distinctions among plant functional types, each of which is allowed to exist over a wide range of aridity values.) Global environmental change then poses two practical questions. First, if aridity changes, on what time scale will it be necessary to update it? Second, will the response to aridity be modified by changes in atmospheric $CO_2$? These are non-trivial questions. We included an extra final paragraph addressing these questions in the Discussion as follows:

"We have developed an empirical soil-moisture stress function that improves the performance of the P model but could also be applied in the context of other models. This research therefore represents a step towards an empirically well-founded representation of the interactions between carbon and water cycling, where the next step would involve the interactive coupling of transpiration and GPP in a land-surface modelling framework. However, we have used a long-term average of climate parameters to calculate the aridity index (AI). Under a changing climate, the AI will change along with changes in vegetation properties such as rooting depth and hydraulic strategy. This poses two practical questions about how to implement our approach under future climate change. First, what is the appropriate timescale at which to update the AI calculation? Second, how will the response to aridity be modified by changes in atmospheric $CO_2$? Both questions are likely related to trait plasticity, plant lifespan, and the speed and magnitude of climate change. Further research is required to address these two crucial issues."

[Figure]

Specific comments:

- L71: while you have given some empirical evidence of photosynthesis levels under drought stress, it should be properly introduced in the introduction section.

As noted in this response above and to the reviewer 1 (comment 5), we have now revised the introduction including more information on this point.

- L103: I think an expanded introduction of the P model is necessary somewhere in the paper.

As mentioned in our response to the first reviewer, both the original P model and the subdaily version have already been published in full. Furthermore, our findings are relevant to all GPP models. We have now revised the text in the abstract, introduction and in the conclusion to emphasize the more general applications of these findings. Please see response to reviewer 1 for the revised text is presented in the response to the second point of the first reviewer.

- L120: so what is the version number of this new model?

We have provided a version number and the full name of the new model (P-model subDaily v1.0.0) in the revised manuscript.

- L360: this concluding paragraph seems quite short and incomplete.

We have provided a new conclusions section in response to comment 9 by reviewer 1.

- Figure 1: perhaps a hexbin plot can show the density of points better than a scatter plot

We thank the reviewer for this suggestion. We have generated a new hexbin plot (see below).

[Figure]

- Figure 5,6: much of them are presenting the same information and can be combined.

We thank the reviewer for this suggestion. We have now combined the two figures. The resulting figure (see below) is now Fig. 6 in the revised manuscript.

[Figure]

The revised caption is:

**Figure 6.** Examples of how the new soil-moisture stress function modifies simulated gross primary production (GPPnew) at nine sites representing the range of climatological aridity compared to how the original stress function, when applied in the sub-daily model, affects simulated GPP (GPPv1.0). The new model is compared to the simulated level of GPP under well-watered conditions (GPPww), and to flux-derived values (GPPobs). Note that the scale varies between the rows. Plots for all the flux tower sites are given in Supplementary Figures 6–8 & 9-11.